Gardnerella species exhibit synergy in their ability to displace Lactobacillus crispatus adhered to HeLa cells

Lima Ângela 1
Castro Joana 2 3
Muzny Christina A. 4
Cerca Nuno nunocerca@ceb.uminho.pt 1 2
1 Laboratory of Research in Biofilms Rosário Oliveira (LIBRO), Centre of Biological Engineering (CEB), University of Minho , Braga , Portugal
2 LABBELS - Associate Laboratory , Braga , Portugal
3 Instituto Nacional de Investigação Agrária e Veterinária (INIAV) , Vairão , Portugal
4 Division of Infectious Diseases, University of Alabama at Birmingham , Birmingham , AL , United States of America
García-Contreras Rodolfo
Electronic publication date: 2025 Nov 5
Publication date: 2025
Volume: 13
Electronic Location ID: e20076
Received 2025 May 8; Accepted 2025 Aug 22
Copyright: ©2025 Lima et al.
Copyright year: 2025
Copyright holder: Lima et al.
License: This is an open access article distributed under the terms of the Creative Commons Attribution License, which permits unrestricted use, distribution, reproduction and adaptation in any medium and for any purpose provided that it is properly attributed. For attribution, the original author(s), title, publication source (PeerJ) and either DOI or URL of the article must be cited.
License URL: https://creativecommons.org/licenses/by/4.0/

Keywords: Bacterial vaginosis, Gardnerella spp., Lactobacillus crispatus, Microbial synergisms, Polymicrobial biofilm, Pathogen competition, Quantitative PCR (qPCR), HeLa cell adhesion assay, Hypothesis testing in microbiota research, Lactobacillus displacement

Funding: Portuguese Foundation for Science and Technology UIDB/04469/2020 2022.13112.BD Associate Laboratory LABBELS LA/P/0029/2020 FCT CEEC National Institute of Allergy and Infectious Diseases (NIAID) R01AI146065-01 This study was supported by the Portuguese Foundation for Science and Technology (FCT) under the scope of the strategic funding of UIDB/04469/2020 unit and the Associate Laboratory LABBELS (LA/P/0029/2020). Ângela Lima is supported by FCT individual PhD fellowship 2022.13112.BD. Joana Castro is supported by FCT CEEC Individual fellowship. Christina A. Muzny is supported by National Institute of Allergy and Infectious Diseases (NIAID) project R01AI146065-01. The funders had no role in study design, data collection and analysis, decision to publish, or preparation of the manuscript.

==============================
Background

Bacterial vaginosis (BV) is the most common vaginal infection in reproductive-age women. It is associated with adverse pregnancy complications, such as preterm birth and low birth weight, in addition to an increased risk of acquisition of HIV and sexually transmitted infections. BV is characterized by a vaginal dysbiosis, involving loss of protective Lactobacillus species (including L. crispatus) and overgrowth of facultative and strict anaerobic bacteria, with Gardnerella species playing a predominant role. However, despite extensive research on BV pathogenesis, its etiology remains unclear, and the sequence of events leading to the displacement of lactobacilli by anaerobic bacteria in women has not yet been fully elucidated. Until 2019, all bacteria belonging to the Gardnerella genus were considered part of the species G. vaginalis. However, it is now recognized that different Gardnerella species exist, each with varying virulence potentials. Recent data have shown that multiple subgroups of Gardnerella spp. are frequently detected simultaneously in the vaginal microbiota of women with BV. With this in mind, we aimed to test the hypothesis that different combinations of known Gardnerella species, isolated from the vaginal microbiota of women with BV, have an enhanced ability to compete against Lactobacillus crispatus, pre-adhered to HeLa cells, thereby facilitating the early stages of BV development.

Methods

Adhesion assays of dual combinations of Gardnerella spp. (G. vaginalis, G. leopoldii, G. swidsinskii, G. piotii) were performed on an in vitro model of HeLa cells, covered with and without L. crispatus. Quantification of the species used in our assays was subsequently performed using quantitative polymerase chain reaction (qPCR).

Results

Our results revealed synergy between different Gardnerella spp., demonstrating their ability to overcome the presumed protective effect of L. crispatus, thereby creating favorable conditions for the development of a polymicrobial biofilm characteristic of BV. The combination of G. vaginalis and G. leopoldii showed the greatest synergistic effect on initial adhesion to HeLa cells while the combination of G. leopoldii and G. swidsinskii had the greatest ability to reduce L. crispatus colonization.

Conclusions

Although this in vitro study does not unequivocally prove that BV is initiated by the disruption of normal vaginal microbiota by Gardnerella spp., it strongly supports this possibility, contributing to a better understanding of BV etiology.

Introduction

Bacterial vaginosis (BV) is the most common vaginal infection in reproductive-age women. It is associated with multiple adverse health outcomes, including infertility, pregnancy complications such as preterm birth and low birth weight (Sethi et al., 2025), in addition to increased risks of HIV and sexually transmitted infection (STI) acquisition, pelvic inflammatory disease, and cervical cancer in the setting of HPV-coinfection (Muzny et al., 2022). Microbiologically, BV is a dysbiosis of the vaginal microbiota, characterized by loss of protective Lactobacillus spp. and an increase in facultative and strict anaerobic bacteria, such as Gardnerella spp., Fannyhessae vaginae, Prevotella spp., and Mycoplasma hominis, among others (Muzny & Sobel, 2023). During BV, a multi-species biofilm is formed on the vaginal epithelium, with Gardnerella spp. thought to be pivotal in the initial stages of biofilm formation (Abou Chacra, Fenollar & Diop, 2022). However, the mechanisms leading to the development of this infection are not fully understood (Redelinghuys et al., 2020). Because BV has a high prevalence and is linked to a high rate of recurrence following treatment, further research regarding its etiology is of utmost importance (Muzny et al., 2022).

We previously hypothesized that BV development is initiated by virulent Gardnerella spp. that displace protective Lactobacillus spp. from the vaginal epithelium. Then, those virulent Gardnerella spp. may initiate the formation of the polymicrobial biofilm (Muzny et al., 2019). Data to support this hypothesis are derived from the observations that Gardnerella spp. have a significantly higher ability to adhere to HeLa cells compared to 29 other BV-associated bacteria (BVAB) (Alves et al., 2014), and that Gardnerella spp. isolated from BV cases have a significantly higher ability to displace lactobacilli pre-adhered to HeLa cells compared to Gardnerella spp. isolated from healthy women (Castro et al., 2013; Castro et al., 2015). In 2019, the description of the Gardnerella genus was emended and three new species, isolated from the vaginal microbiota, were proposed in addition to the previously identified Gardnerella vaginalis: Gardnerella leopoldii, Gardnerella piotii, and Gardnerella swidsinskii (Vaneechoutte et al., 2019). Although multiple subgroups of Gardnerella spp. are frequently detected simultaneously in the vaginal microbiota (Hilbert et al., 2017), there is no research on their interactions when adhering to vaginal epithelial cells. With this in mind, we hypothesized that frequent vaginal co-colonization by different Gardnerella spp. is the result of synergistic interactions that occur in the early stages of BV development, namely, in the initial adhesion to the vaginal epithelium. To test our hypothesis, we devised an experimental model, where pair-wise combinations of G. vaginalis, G. piotii, G. leopoldii and G. swidsinskii challenged pre-condition HeLa cells colonized by Lactobacillus crispatus, a key indicator of vaginal health (Huang et al., 2014).

Materials & Methods

Growth conditions and strains

G. vaginalis UM 121, G. leopoldii UGent 09.48, and G. swidsinskii CCUG 44005 were isolated from women with BV, G. piotii UGent 18.01 was isolated from a woman with intermediate microbiota (Nugent score 4–6), and L. crispatus CCUG 44128 was isolated from a healthy woman. L. crispatus, G. vaginalis, G. piotii, G. leopoldii, and G. swidsinskii were grown on Columbia blood agar base (CBA) (Oxoid, Basingstoke, UK) supplemented with 5% (v/v) defibrinated horse blood (Oxoid), at 37 °C under 10% CO2. Escherichia coli ATCC 25922 was also grown on CBA, but in an aerobic atmosphere at 37 °C. The inocula were cultured in New York City III broth supplemented with 10% (v/v) inactivated horse serum (Biowest, Nuaill’e, France) for 48 h at 37 °C, under 10% CO2 in static conditions (Castro et al., 2015). The optical densities used for all Gardnerella spp. was 0.18 which corresponds to a bacterial concentration of 1  × 108 CFU/mL, and for L. crispatus was 0.13 that corresponds to 5 × 107 CFU/mL, as previously demonstrated (Rosca et al., 2022; Lima et al., 2024).

Adhesion assays

Human cervical HeLa cells (from the American Tissue Culture Collection, ATCC CCL-2) were cultured in Dulbecco’s Medium (Pan Biotech, Aidenbach, Germany) supplemented with 15% (v/v) Fetal Bovine Serum (FBS; Pan Biotech) and 1% (v/v) antibiotic (ZellShield®, Berlin, Germany) at 37 °C and in 5% CO2. Cells were cultured in 24-well tissue culture plates (Orange Scientific, Braine L’Alleud, Belgium) for 48 h. Before the displacement and adhesion assays, the confluence of the cells was verified with an inverted microscope (Leica DM IL). Next, the HeLa cells were washed with one mL of 1 × PBS to remove the non-adherent cells and culture medium, as previously described (Castro et al., 2013). The capacity of dual combinations of Gardnerella spp. to displace L. crispatus pre-adhered to HeLa cells and to also adhere to HeLa cells was evaluated using a previously developed protocol, with several modifications (Castro et al., 2013). After washing the 24-well tissue culture plates, one mL of L. crispatus at 5 × 107 CFU/mL was added to each of the wells. After 4 h of incubation the media was removed, and the wells were washed with one mL of 1 × PBS. Then L. crispatus was challenged with 500 µL of each of the four Gardnerella spp. at 1  × 108 CFU/mL, in dual combinations. These mixtures were also added to the HeLa cells without L. crispatus, and the plates were incubated for 30 min. After incubation, all the wells were washed with one mL of 1 × PBS. To detach all of the cell culture, including the adherent bacteria, 100 µL of Trypsin-EDTA (Sigma-Aldrich) was added and the well plates were incubated for 5 min at 37 °C and in 5% CO2. Next, 900 µL of PBS 1 × was added to the wells to stop the trypsinization. All of the wash steps were uniformly performed across the assays. Two controls were included. To determine the ability of each Gardnerella spp. to adhere to the HeLa cells, one mL of each species at 1 × 108 CFU/mL was added to each well containing only HeLa cells. To determine the ability of each Gardnerella spp. to displace pre-adhered L. crispatus, one mL of each species at 1 × 108 CFU/mL was added to wells containing L. crispatus adhered to the HeLa cells, prepared in the same way as described for the test assays. All assays were performed three independent times, with technical duplicates.

Construction of calibration curves for quantitative Polymerase chain reaction (qPCR)

The calibration curves for quantitative polymerase chain reaction (qPCR) were constructed, as previously described (Lameira et al., 2024). The biomass of each strain was resuspended in 1 × PBS. The L. crispatus suspension was sonicated (Ultrasonic processor Cole Parmer) three times for 10 s at 40% with 10 s intervals, on ice, between cycles. The optical density (OD) at 620 nm of each suspension was adjusted to a concentration of 1 × 108 CFU/mL, and serial dilutions ranging from 108 to 104 were performed. 100 µL of E. coli at 1 × 108 CFU/mL was added to each dilution of target bacteria, as an exogenous control, to normalize extraction efficiency. In addition, 100 µL of HeLa cells suspension was also added to each dilution. The HeLa cells suspension was added to each dilution of the calibration curves to account for any potential loss of efficiency in the extraction and quantification process that may have occurred with the adhesion assay samples. Each suspension was centrifuged at 16,000 g for 10 min at 4 °C. The supernatant was discarded, and the pellet was frozen at −20 °C overnight for further gDNA extraction.

Genomic DNA (gDNA) extraction and bacterial quantification by qPCR

gDNA was extracted from pellets using a commercial Blood & Tissue Kit (Qiagen, Hilden, Germany), following the manufacturer’s instructions, with some modifications. Briefly, the pellets were re-suspended in 300 µL of 1 × PBS and 300 µL of buffer AL was added to the mixture. In the final step, DNA was eluted in DNAse free water. Genomic DNA concentration was measured using the Nanodrop ND-One-W (Thermo Scientific). Bacterial quantification was performed by qPCR, as previously described (Lameira et al., 2024). All runs were performed in a CFX96 thermal cycler (Bio-Rad, Hercules, CA, USA), using the following cycle parameters: 95 °C for 3 min, followed by 40 cycles of 95 °C for 5 s and 60 °C for 20 s. The qPCR amplifications were conducted in 10 µL reaction mixtures containing five µL of Xpert Fast SYBR Master Mix (Grisp, Porto, Portugal), one µL of primer mixture (10 µM), two µL of DNase-free water, and two µL of a 1:40 dilution of extracted gDNA. The software calculated the cycle threshold (Ct), with the baseline threshold being established using an inter-run calibrator in every run. Any Ct value ≤35 was considered positive. Normalization of the qPCR results was achieved through the equation, where ΔCt corresponds to Δ cycle threshold. The target species corresponded to G. vaginalis, G. piotii, G. leopoldii, G. swidsinskii, and L. crispatus and the control species corresponded to E. coli. Calibration curves were generated using the above equation, with ΔCt values plotted on the y-axis and known bacterial concentrations (CFU/mL) on the x-axis. The bacterial concentration was determined using the generated calibration curves, with the formula, where corresponds to the bacterial concentration and A and B are values given by the generated calibration curve. Primer efficiency (E) was determined from the slope of a standard curve, constructed with 10-fold serial dilutions of each target bacteria, E = [10(−1/(slope)) –1] × 100 (Lameira et al., 2024). The qPCR primers used are presented in Table 1. All sets of primers used were obtained from other studies (Collins et al., 2018; Ribeiro et al., 2019; Latka et al., 2022). The GenBank accession numbers for the genes used to design the primers were 45576854 (G. vaginalis), 97755823 (G. leopoldii), 95681794 (G. swidsinskii), V00296.1 (E. coli), and LC065039.1 (L. crispatus). There was no information on the gene ID for the G. piotii DNAg gene, but the accession number of the strain used to design its primers was CP180510.1. The qPCR runs were done three independent times, with technical duplicates. The main steps of the experimental protocol are depicted in Fig. S1.

Table 1 Primers sequence.

Sequence of the primers used in this study, with relevant associated data.

Species	Target gene	Primer sequence (5′–3′)	GC content %	Amplicon size (bp)	Melting temperature (°C)	Reference	
G. vaginalis	DNAg	TATTATAACTAAAGCTGCTG	30.0	141	81.5	Latka et al. (2022)	
TCGCCACTATAGTCG	53.3	
G. piotii	DNAg	AGCTGCTTACGATTATAGT	36.8	162	82	Latka et al. (2022)	
TTACTCATTCTAAGCTTAATAG	27.3	
G. leopoldii	DNAg	GATACTGCACTGTATCGA	44.4	139	79.5	Latka et al. (2022)	
CAGTATCAATACCAGCC	46.1	
G. swidsinskii	DNAg	ATTTAGTTAGATATTTGGCAA	23.8	95	81	Latka et al. (2022)	
ATAGTCATATATTCCGCGC	42.1	
L. crispatus	16S rRNA	AGCGAGCGGAACTAACAGATTTAC	30.0	154	83.5	Collins et al. (2018)	
AGCTGATCATGCGATCTGCTT	53.3	
E. coli	lacZ	AGCGAAACCGCCAAGACTGTTA	50.0	135	86	Ribeiro et al. (2019)	
AGCGAGCGGAACTAACAGATTTAC	45.8	

Analytical sensitivity and specificity

The limit of detection was determined when the calibration curves were constructed, as mentioned above. The lowest DNA concentration of each curve was the limit of detection of our assays. Regarding specificity, a BLAST search was performed for every pair of primers and each bacterium used in this study, to determine if the primers detected only their specific target. The analytical specificity was also determined by the melting curve of the qPCR results.

Inhibition testing

To determine the presence of qPCR inhibitors, serial ten-fold dilutions of one gDNA sample from the assays were performed with DNase free water. Then, a qPCR was carried out, targeting the E. coli exogenous control. The software calculated the efficiency of the qPCR reaction.

Statistical analysis

The results obtained from the displacement assays were analyzed using the multiple t-test (Holm-Sidak) and the results obtained from the adhesion of Gardnerella spp. to the HeLa cells were analyzed using the t-test for paired samples. GraphPad Prism 2016 was the software used. A p-value ≤ 0.05 was considered statistically significant.

Results

Calibration curves

We started by constructing qPCR calibration curves, depicted in Fig. 1, to quantify the different Gardnerella spp. and L. crispatus in our adhesion assays. As shown in Fig. 1, all calibration curves presented a linearity of (R2) of ≥ 0.98.

Figure 1 Calibration curves to relate cycle threshold detection (Δ Ct) and bacterial load quantification.

Each dot represents the average of three independent assays and the standard deviation.

Adhesion of pairwise combinations of Gardnerella spp. on HeLa cells

To evaluate the ability of the selected Gardnerella spp. to adhere to and colonize HeLa cells, and to explore potential synergistic effects in the initial adhesion process, we quantified their adhesion in single and pairwise combinations after a 30-min challenge. As shown in Fig. 2, a variable synergistic effect was observed, with most of the combinations tested resulting in significantly (p ≤ 0.05) higher adhesion numbers in the dual-species challenges compared to the single-species challenges. The exception was G. piotii cultured with G. leopoldii, whereby, their combination resulted in a total bacterial colonization lower than the G. leopoldii colonization alone. The combination of G. vaginalis and G. leopoldii showed the highest synergistic effect on the initial adhesion to HeLa cells, as it was 13 times more efficient than the G. vaginalis and G. piotii pair (the least efficient dual-species combination). The significant statistical differences observed in this experiment are highlighted in Table S1.

Figure 2 Adhesion of dual combinations of Gardnerella spp. to HeLa cells.

The bars represent the Gardnerella spp. concentration alone or in dual combinations that adhered to HeLa cells. The bars also represent the mean and the standard deviation of three independent experiments. An asterisk (*) represents statistical significance between Gardnerella spp. alone; δ represents statistical significance between G. vaginalis (Gv) alone and the mix; λ represents statistical significance between G. piotii (Gp) alone and the mix; ɛ represents statistical significance between G. leopoldii (Gl) alone and the mix; α represents statistical significance between G. swidsinskii (Gs) alone and the mix; Φ represents statistical significance between Gv alone and Gv in the mix and β represents statistical significance between Gs alone and Gs in the mix (t-test, paired samples, p ≤ 0.05). Gv, G. vaginalis; Gp, G. piotii, Gl, G. leopoldii and Gs, G. swidsinskii.

Displacement of L. crispatus by pairwise combinations of Gardnerella spp.

After assessing the potential synergistic effects between the different Gardnerella spp., we performed a second experiment, in which L. crispatus was first allowed to colonize HeLa cells for 4 h, after which the single and dual-species challenges were repeated. This experiment aimed to simulate a possible external challenge to the vaginal epithelium of a healthy woman colonized by L. crispatus. The main objective was to observe if any synergistic effects could be found when attempting to displace the pre-adhered L. crispatus. As shown in Fig. 3, most of the dual-species combinations of Gardnerella spp. tested resulted in a higher ability to displace the pre-adhered L. crispatus, in comparison with the single-species challenges. Also, it was observed that G. leopoldii was the tested species with the highest innate ability to displace L. crispatus from the HeLa cells. Interestingly, the observed effect was combination-dependent, with the combination of G. leopoldii and G. swidsinskii having the highest ability to reduce L. crispatus colonization. Compared to the less efficient pair G. piotii and G. swidsinskii, the G. leopoldii and G. swidsinskii pair was 10 times more efficient in displacing L. crispatus. The significant statistical differences observed in this experiment are highlighted in Table S2.

Figure 3 Displacement of L. crispatus. by dual combinations of Gardnerella spp.

The bars represent L. crispatus concentrations before (Control) and after being challenged with Gardnerella spp. The bars also represent the mean and the standard deviation of three independent experiments. An asterisk (*) represents statistical significance (multiple t-test, p ≤ 0.05). Gv, G. vaginalis; Gp, G. piotii, Gl, G. leopoldii and Gs, G. swidsinskii.

Adhesion of pairwise combinations of Gardnerella spp. on HeLa cells covered with L. crispatus

Parallel to determining L. crispatus quantification after the Gardnerella spp. challenges, we also quantified how many Gardnerella spp. were able to remain adhered to the HeLa cells after the challenges. Not surprisingly, when comparing Fig. 4 with Fig. 2, we verified that in most cases, in the presence of L. crispatus, the Gardnerella spp. pairs concentration was significantly lower than in its absence. The only exception was the dual culture of G. vaginalis and G. swidsinskii, suggesting that unique interactions between these two species may allow them to overcome L. crispatus defenses more effectively. Interestingly, the G. piotii-G. leopoldii pair adhered less than G. leopoldii alone, although the difference was not significant. In contrast, the combination of G. vaginalis and G. swidsinskii showed the highest ability to adhere to HeLa cells in the presence of L. crispatus. The significant statistical differences observed in this experiment are highlighted in Table S3.

Figure 4 Gardnerella spp. dual combination adhesion to HeLa cells with pre-adhered L. crispatus.

The bars represent the Gardnerella spp. concentration alone or in dual combinations that adhered to HeLa cells. The bars also represent the mean and the standard deviation of three independent experiments. An asterisk (*) represents statistical significance between Gardnerella spp. alone; δ represents statistical significance between Gv alone and the mix; λ represents statistical significance between Gp alone and the mix; θ represents statistical significance between Gp alone and Gp in the mix; ɛ represents statistical significance between Gl alone and Gl in the mix; α represents statistical significance between Gs alone and Gs in the mix and β represents statistical significance between Gs alone and Gs in the mix (t-test, paired samples, p ≤ 0.05). Gv, G. vaginalis; Gp, G. piotii, Gl, G. leopoldii and Gs, G. swidsinskii.

Discussion

Despite many years of extensive research on the pathogenesis of BV, its etiology is not yet fully determined, hindering the development of improved diagnostic, treatment, and prevention measures (Redelinghuys et al., 2020). As such, continued investment in the study of BV pathogenesis is essential. Here we investigated the interactions of four Gardnerella spp. (in dual combinations) on their ability to adhere to HeLa cells and also displace pre-adhered L. crispatus. L. crispatus is frequently used as a model to study BV because it is the dominant protective vaginal Lactobacillus spp. in most women of childbearing age, harboring antimicrobial properties (Witkin & Linhares, 2017). Furthermore, women with a vaginal microbiota dominated by L. crispatus are generally at lower risk of developing BV, and its presence during BV treatment is also associated with better outcomes (Chee, Chew & Than, 2020). Previous studies have demonstrated that Gardnerella spp. isolated from BV cases are able to significantly displace L. crispatus (Castro et al., 2013; Castro et al., 2015). Interestingly, in this study, when we tested dual-species Gardnerella combinations, we observed synergism in most cases. However, the combination that showed the greatest ability to displace L. crispatus was not the same pair that demonstrated the greatest capacity to adhere to HeLa cells. This suggests that the ability of Gardnerella spp. to displace L. crispatus may not be solely dependent on their capacity to adhere to epithelial cells, but also on other factors, such as competitive exclusion (Ghoul & Mitri, 2016) or modulation of host immune responses (Verspecht et al., 2021). It is important to highlight that our adhesion assay measured post-wash recovery of bacteria, and that this readout reflected the sum of all adhesion forces (electrostatic, hydrogen bonding, van der Waals, hydrophobic, and hydrophilic interactions) (Elbourne et al., 2019). These assays reported the net number of bacteria remaining after a defined wash procedure, and therefore cannot intrinsically distinguish between direct competition for binding sites on the epithelial monolayer or weakened adhesion forces leading to differential wash-off from the plastic, which is a technical limitation (Song, Koo & Ren, 2015). However, it is important to emphasize that all experiments were subjected to identical wash stringencies. Therefore, any differences in L. crispatus recovery must have arisen from changes in its adhesion strength induced by the presence (and specific composition) of the Gardnerella spp., rather than by variable wash conditions.

Our experiments support the feasibility of our hypothesis and are consistent with data presented in other studies demonstrating the potential synergism of Gardnerella spp. in vivo, with BV-positive participants having multiple species of Gardnerella detected in their vaginal specimens (Balashov et al., 2014; Munch et al., 2024). The fact that co-existence of multiple Gardnerella spp. may facilitate BV pathogenesis can be explained by significant differences in the accessory genome of these species, which can result in complementary virulence (Vaneechoutte et al., 2019). Nevertheless, it has been reported that antagonistic interactions can occur between different Gardnerella spp., wherein reduced growth rates were observed (Khan, Voordouw & Hill, 2019). While this prior study shows different ecological interactions to the ones reported here, it should be highlighted that both studies are fundamentally different in their methodologies. The conditions we tested did not include bacterial growth in rich media, a condition very different from what occurs in the vaginal niche.

Another limitation of our study was the use of HeLa cells, a tumoral cell line, instead of a normal epithelial cell line. We used HeLa cells in our experiments as they have fast growth, good adherence, and high viability, ideal characteristics for repetitive and reproducible assays. Because they are a stable and well-characterized line, consistent results between different laboratories can be ensured (Lih Yuan et al., 2023). In addition, we wanted to compare these results with our previous studies (Castro et al., 2013; Castro et al., 2015). However, we are aware that HeLa cells have reduced expression of certain receptors, namely, Toll-like receptors, which are essential for the recognition of microorganisms and the initiation of a host inflammatory response, when compared to normal epithelial cells (Zhao et al., 2023). Furthermore, our study was limited by the use of pair-wise interactions, the use of one strain per species, and a single methodological approach. While we acknowledge that strain-to-strain variation may exist, our goal was to test an initial specific hypothesis. Indeed, we have demonstrated that synergistic interactions between different Gardnerella spp. provide a plausible explanation for the in vivo co-colonization of multiple Gardnerella spp., supporting our initial hypothesis.

Conclusions

Overall, our findings support the notion that different Gardnerella spp. contribute to successful initial colonization of the vaginal epithelium, highlighting the critical role of Gardnerella spp. as early colonizers in the pathogenesis of BV. We do not claim direct clinical extrapolation but instead propose a plausible mechanism, namely cooperative adhesion among Gardnerella spp., that may help to explain in vivo patterns observed in BV. Nevertheless, our results also support the notion that L. crispatus can have a protective effect against Gardnerella spp., since in most of the dual-species combinations tested, there was a decrease in Gardnerella spp. concentration in its presence. However, there may be a breaking point wherein the concentration of L. crispatus is no longer able to prevent Gardnerella spp. overgrowth, leading to the onset of BV. This is of relevant importance since the initial adhesion to epithelial cells is a crucial step in BV development. A further understanding of the dynamics of this breaking point is crucial for developing novel treatment and prevention strategies aimed at restoring and maintaining a healthy vaginal microbiota.

Supplemental Information

Supplemental Information 1 MIQE guidelines checklist

Supplemental Information 2 Supplemental figure and tables

Supplemental Information 3 Raw Data

Values of cycle threshold of each sample used in this paper, for different primers (species). With those values we constructed the calibration curves and also the concentration of each species after the adhesion assays. The concentration values allowed us to statistically analyze the observed differences.

Supplemental Information 4 Publication license

We would like to thank Carlos Costa for creating Fig. S1.

Additional Information and Declarations

Competing Interests

Author Contributions

Data Availability

Christina Muzny reports receiving grants to her institution from Abbott, BioNTech, and Gilead Sciences, Inc. She also reports honorarium and/or consulting fees from Abbott, BioNTech, bioMerieux, Cepheid, Elsevier, Merck Manuals, UpToDate, and Roche. The other authors have no disclosures.

Ângela Lima performed the experiments, analyzed the data, prepared figures and/or tables, authored or reviewed drafts of the article, and approved the final draft.

Joana Castro analyzed the data, authored or reviewed drafts of the article, and approved the final draft.

Christina A. Muzny conceived and designed the experiments, authored or reviewed drafts of the article, and approved the final draft.

Nuno Cerca conceived and designed the experiments, analyzed the data, authored or reviewed drafts of the article, and approved the final draft.

The following information was supplied regarding data availability:

Data is available in the Supplemental Files.

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
