# Peer review of "Gardnerella species exhibit synergy in their ability to displace Lactobacillus crispatus adhered to HeLa cells"

_PeerJ, doi:10.7717/peerj.20076_

## Round 0.1 · original submission · Major Revisions

Please address all reviewer's comments.

Reviewer 1 ·

Basic reporting

- The manuscript is generally written in clear and professional English.
- A few grammatical issues and phrasing inconsistencies were observed (e.g., vague expressions and awkward constructs).
- The figures and tables are relevant, appropriately labeled, and informative.
- References are current and support the context well.

Experimental design

- The research question is relevant and addresses a meaningful gap in understanding the pathogenesis of bacterial vaginosis.
- Methods are described with sufficient detail, including clear qPCR protocols and statistical approaches.
- The study used an appropriate in vitro model (HeLa cells), though use of primary vaginal epithelial cells could strengthen future studies.

Validity of the findings

- Results are statistically sound and based on triplicate experiments.
- Key findings are supported by the data, though some claims could benefit from more cautious language or numerical clarification.
- Limitations are adequately acknowledged by the authors.

Additional comments

- Line 45: 'associated with various pregnancy complications' is vague – consider specifying.
- Line 58: 'with and without L. crispatus previously adhered – rephrase for clarity.
- Line 141: Consider clarifying which previous study is referred to.
- Line 224: '13 times more efficient' – consider providing statistical values.
- Line 249: 'combination-dependent' – clarify the specific combinations.
- Line 325: Typo in 'dual species’ combinations – correct to 'dual-species combinations'.

- This is a strong study with well-supported conclusions and excellent methodological detail.
- The study makes a valuable contribution to our understanding of microbial interactions in BV pathogenesis.
- A few language improvements and clarification of some claims would enhance the overall quality.
-It is preferable to write the keywords
-References should preferably be written as numbers in the text of the research, as in the following example: [1 ], [2 ], etc…
- As for the references, write them correctly, but the years must be placed in brackets and the scientific names must be written in italics. I indicated all of that in yellow.

Annotated reviews are not available for download in order to protect the identity of reviewers who chose to remain anonymous.

·

Basic reporting

The experiment is of interest to BV research, as it provides insights into the synergistic interactions among different Gardnerella species in displacing lactobacilli adhesion in cell lines. The language used in this article is generally OK. However, there are still some ambiguities that the authors should clarify. In particular, certain sections of the paper appear to use the terms epithelial and HeLa interchangeably, which I believe is inappropriate. HeLa cells, although originally derived from cervical epithelial cells, are tumor cells and may not behave in the same way as normal epithelial cells. Also, scientific names/genera are not properly italicized in places.

Experimental design

Its main aim is to determine the synergistic effect of different Gardnerella species in replacing Lactobacillus crispatus adhesion to HeLa cells.

However, the study relies on a single experimental approach to assess bacterial adhesion and persistence—namely, the adhesion assay followed by qPCR quantification.

While the results suggest a potential trend in which L. crispatus may be displaced by Gardnerella, further experiments are necessary to substantiate this finding. Although the data are intriguing, the work remains very preliminary. Several aspects require clarification to strengthen the study.

Major points:

1) How long were the HeLa cells cultured prior to the experiment? What was the rationale behind choosing this specific time point?

2) Have the authors considered the biological differences between HeLa cells and normal epithelial cells? Would the results be consistent if a normal epithelial cell line were used instead?

3) Is it possible that L. crispatus adhered to the plastic surface of the culture wells and was differentially washed out depending on the Gardnerella mixture used?

4) Why did the authors choose a 4-hour time point for Lactobacillus colonization? Is this duration sufficient for effective bacterial adherence and colonization?

5) The results only demonstrate that the addition of a Gardnerella mixture may enhance adherence to HeLa cells and potentially displace L. crispatus. However, they do not provide any evidence regarding biofilm formation.

6) Were the Gardnerella strains used in the study associated with BV or non-BV conditions? Would the outcomes differ if the experiments were conducted with strains isolated from BV-positive versus BV-negative cases?

Validity of the findings

In my opinion, this work is still very preliminary. Additional control experiments are necessary to support more conclusive interpretations. The current interpretation appears somewhat overstated. I would recommend a more cautious approach, stating that the Gardnerella mixture may alter the adhesion of L. crispatus. Further experiments should also include other bacteria associated with BV, as well as non-BV Gardnerella strains, before drawing such conclusions. Alternative methods, such as Biacore analysis or fluorescence microscopy, should be employed to validate the findings.

Reviewer 3 ·

Basic reporting

The background and scientific rationale for the study are well-reasoned. The hypothesis that multiple species of Gardnerella may cooperate to facilitate cell colonization and growth is worthy of study.

There are deficits in reporting, including:
1. A sonication step is described without describing the sonicator. “The L. crispatus suspension was sonicated 3 times for 10 s at 40% with 10 s intervals, on ice, between cycles.” This is not reproducible by another investigator.

2. “100 µL of E. coli at 1 Å~ 108 CFU/mL was added to each dilution of target bacteria, as an exogenous control, to normalize extraction efficiency.” How does this normalize extraction efficiency?

3. It is unclear what is being assayed by qPCR. Is it the entire cell culture well, including HeLa cells, at the end of the experiment, after washing? If so, how was this harvested? Was the wash solution assayed for non-adherent bacteria? If not, why not have an assessment of the fraction of adhered vs. non-adhered bacteria?

Experimental design

The authors chose to use HeLa cells for experiments measuring the ability of bacteria to adhere to human cells. HeLa cells were originally derived from cervical cancer tissue and are not vaginal epithelial cells. Furthermore, they have high aneuploidy and with these multiple chromosomal abnormalities may not represent a relevant model where binding reflects what transpires in the human vagina. A justification for this decision is needed. For example, if the cell surface of HeLa cells is very different from normal human vaginal epithelial cells then the adherence findings described here may be irrelevant.

Validity of the findings

Overall, the authors provide evidence that combinations of Gardnerella species are more effective in attaching to HeLa cells and displacing Lactobacillus crispatus from HeLa cells compared to single species of Gardnerella. This suggests some degree of cooperation and lack of antagonism among Gardnerella species in this model.

Additional comments

The findings are interesting, and the hypothesis is well considered. It is unclear how this model using HeLa cells reflects what is found in the human vagina.

---

## Round 0.2 · Minor Revisions

Please perform a last round of careful proofreading to eliminate typos or minor mistakes as suggested by the reviewers.

Reviewer 1 ·

Basic reporting

• In lines 241–250, the displacement analysis of Lactobacillus crispatus by Gardnerella species is based on a HeLa cell model. However, Khan et al. (2019) showed that Gardnerella interactions can also be antagonistic, not solely synergistic. It would strengthen the manuscript to discuss this apparent discrepancy with previous findings.
• While the authors justify the use of HeLa cells by referencing earlier studies (e.g., Castro et al., 2013), more recent work (Zhao et al., 2023) highlights significant limitations of HeLa cells, such as reduced expression of innate immune receptors. This limitation should be more clearly emphasized in the discussion.
• The manuscript is generally well-written, but some sentences are overly complex or may be difficult for a broader international audience.
• Suggested edits:
o Line 77: “due to HPV co-infection” could be rephrased for clarity as: “which may be exacerbated by HPV co-infection.”
o Line 85: “virulent Gardnerella spp. that displace protective Lactobacillus spp.” could be simplified to avoid ambiguity.
• I recommend that the authors seek help from a colleague fluent in academic English or consider using a professional language-editing service to further refine the clarity and flow of the text.

Experimental design

• The study addresses a timely and important topic: the early steps in bacterial vaginosis (BV) pathogenesis.
• The use of qPCR-based quantification strengthens the reliability of the results.
• The manuscript is generally well-structured, and the experimental design is clear and replicable.
• The findings offer valuable mechanistic insights into species-level interactions among Gardnerella spp., contributing to the evolving understanding of BV onset

Validity of the findings

1. HeLa Cell Model Limitation: A more thorough discussion of the limitations of using HeLa cells instead of normal epithelial cells is warranted.
2. Contextualizing Synergy Findings: The synergistic effects reported should be critically compared with studies reporting competition among Gardnerella strains (e.g., Khan et al., 2019).
3. Lack of Complementary Assays: The study relies solely on adhesion assays; incorporation of imaging or host-response assays would have further validated the results.
4. Language and Style Issues: While minor, improving the English in several areas would enhance readability.
5 I appreciate that the authors have made the raw data available, which significantly enhances the reproducibility of the work. However, the supplementary datasets would be more useful to future readers if accompanied by more descriptive metadata (e.g., strain origin, growth conditions, inoculum density).
6 While the results are compelling, the data interpretation would benefit from a more balanced discussion that includes potential methodological constraints (e.g., artificial nature of HeLa monolayers).

Additional comments

• Introduction (Lines 73–104): The introduction could be improved by further elaborating on the knowledge gap this study aims to fill. I recommend including a brief synthesis of recent reviews (e.g., Muzny et al., 2022) to highlight the novelty and necessity of the current work.
• Methods (Lines 105–205): The methods are described in sufficient detail. However, summarizing the experimental workflow in a supplementary table or figure would help facilitate replication.
• Results Section: The figures are clear, but the narrative would benefit from integrating a comparative summary table that statistically contrasts all pairwise combinations

Annotated reviews are not available for download in order to protect the identity of reviewers who chose to remain anonymous.

·

Basic reporting

The manuscript is much improved as it, at this point, addressed most of the concerns and stated the limitation of the experiments. However, there are still some typos and formatting issues, for example, Gardnerella spp. in line 142 should be italicized. The authors should carefully and rigorously revisit the manuscript.

Experimental design

The experiment is still preliminary as the additional experiment cannot be conducted to further substantiate the hypothesis due to financial issues. In any case, the authors had carefully disclaimed and discussed the potential limitation of the experiments already in the revised manuscript.

Validity of the findings

Despite the preliminary nature, the manuscript should be of interest to researchers who are working in the BV field.

---

## Round 0.3 · accepted · Accept

Thanks for addressing all comments!

·

Basic reporting

The authors have added the statement that helps improve the clarity of the work.

Experimental design

Despite the preliminary nature of the work, the body of the work is still interesting and should be of interest to the researchers interested in infection mechanism of BV

Validity of the findings

The work might still have some limitation for interpreting the data.